# Three-dimensional open architecture enabling salt-rejection solar evaporators with boosted water production efficiency

Kaijie Yang [1,3], Tingting Pan[1,3], Saichao Dang[2], Qiaoqiang Gan [2] ✉ & Yu Han [1] ✉

Direct solar desalination exhibits considerable potential for alleviating the global freshwater crisis. However, the prevention of salt accumulation while maintaining high water production remains an important challenge that limits its practical applications because the methods currently employed for achieving rapid salt backflow usually result in considerable heat loss. Herein, we fabricate a solar evaporator featuring vertically aligned mass transfer bridges for water transport and salt backflow. The 3D open architecture constructed using mass transfer bridges enables the evaporator to efficiently utilize the conductive heat that would otherwise be lost, significantly improving the water evaporation efficiency without compromising on salt rejection. The fabricated evaporator can treat salt water with more than 10% salinity. Moreover, it can continuously and steadily work in a real environment under natural sunlight with a practical solar-to-water collection efficiency of >40%. Using the discharged water from reverse osmosis plants and sea water from the Red Sea, the evaporator demonstrates a daily freshwater generation rate of ~5 L/m², which is sufficient to satisfy individual drinking water requirements. With strong salt rejection, high energy efficiency, and simple scalability, the 3D evaporator has considerable promise for freshwater supply for water-stressed and off-grid communities.

As a sustainable desalination technology, direct solar desalination has the advantages of low cost, off-grid capability, and zero carbon footprint, and is particularly useful for remote areas and distributed communities[1–13]. The solar-to-vapour conversion efficiency for single-stage processes has approached 100% in the past few years. The evaporation rate can even surpass the theoretical limit for exclusive solar-driven evaporation by exploiting environmental heat[14–16]. However, high evaporation rates cannot be maintained during the evaporation of saline water (e.g., seawater and concentrated brine discharged from reverse osmosis (RO) facilities) because of salt accumulation[17–20].

For instance, commercial solar stills (Aquamate Solar Still®) cannot work for practical desalination applications because their evaporators cannot be replaced or cleaned, thus resulting in a short lifespan[21]. Recently, innovative strategies, which can be classified into two general categories, (1) hydrophobic light-absorbing layer design[10,19,22] and (2) fluid convection enhancement[23–26], were developed to address this challenge. For example, Janus structures having an upper hydrophobic solar-absorber layer and an underlying hydrophilic water-absorbing layer were proposed[22]. In this design, saline water cannot reach the upper surface because of its hydrophobicity, thereby preventing

[1]Advanced Membranes and Porous Materials (AMPM) Center, Physical Sciences and Engineering Division, King Abdullah University of Science and Technology (KAUST), Thuwal, Saudi Arabia. [2]Materials Science Engineering Program, Physical Science and Engineering Division, King Abdullah University of Science and Technology (KAUST), Thuwal, Saudi Arabia. [3]These authors contributed equally: Kaijie Yang, Tingting Pan. ✉ e-mail: qiaoqiang.gan@kaust.edu.sa; yu.han@kaust.edu.sa

surface salt accumulation. However, despite the excellent salt rejection capability associated with this design, its energy conversion efficiency is limited by the rapid heat dissipation from the solar absorber to the bulk water underneath. Likewise, although salt rejection can be realized by improving the fluid convection between the evaporation surface and bulk water, fluid exchange not only removes salt but also takes heat away from the evaporation surface, thereby resulting in a relatively low vapour generation rate[3]. Thus, the paradox between salt rejection and heat loss remains one of the most challenging barriers faced by solar-driven interfacial evaporation strategies.

Herein, we present a rationally designed 3D salt-rejection evaporator that achieves stable and efficient water evaporation by enhancing the salt backflow and conductive heat recovery. The key component of the evaporator is a number of vertically aligned mass transfer bridges (MTBs) containing abundant hydrophilic microchannels. In addition to facilitating salt and water transport, MTBs separate the solar absorber from bulk water, thus creating a highly open 3D space through which additional water can be evaporated by conductive heat from the solar absorber. The solar-driven vapour generation rate and practical water production performance of this 3D evaporator are evaluated under laboratory and outdoor conditions. Our experimental results demonstrate that the 3D evaporator can stably and continuously operate without salt accumulation when processing high-salinity water (e.g., 12–14 wt.% NaCl solution and concentrated brine from RO facilities) while achieving a high vapour generation rate of ~1.64 kg/m²/h. A scaled-up solar evaporator is tested on a rooftop and Red Sea to demonstrate its practical applications. Moreover, a daily water collection rate of ~5 L/m² with a practical solar-water collection efficiency of >40% was demonstrated, better than a previously reported record of ~22%[25]. This 3D solar evaporator promises to increase access to affordable freshwater in water-stressed areas and improve our resilience to natural disasters in future.

## Results

### Evaporation structure design and fabrication

For conventional salt-rejection solar evaporation systems, water evaporation is confined to the solar absorber surface, and the salt backflow is accompanied by an undesired heat dissipation from the solar absorber to bulk water, thus resulting in a low evaporation rate. This limitation can be solved to a considerable extent by our 3D evaporator. As illustrated in Fig. 1a, the top surface of our evaporator is a solar absorber layer used for light-to-heat conversion to generate vapour. Beneath the solar absorber are a number of vertically aligned MTBs connecting the saline water to the solar absorber. MTBs have hydrophilic microchannels that can pump saline water to the solar absorber via a capillary force. Furthermore, excessive salt can flow back into the bulk water through these brine-filled microchannels via diffusion and convection (Fig. 1b-1). The adequate mass transfer via a high density of MTBs ensures a continuous water supply and an efficient salt backflow, thus enabling a unique salt rejection capability. Unlike conventional salt-rejection systems, where the heat conducted from the solar absorber to the bulk water is simply dissipated and considered "wasted," the MTBs can efficiently recover this conductive heat to generate additional vapour from the brine flowing through their microchannels (Fig. 1b-2). The microchannels within the MTBs and macrochannels between the spaced MTBs together form a highly open structure that allows the generated vapour to be easily released from the MTB surfaces in all directions. We envision that by optimizing the MTB height, conductive heat can be largely confined in them for vapour generation, thereby significantly improving the water evaporation efficiency.

We achieved the designed structure by fabricating the top solar absorber layer by loading carbon nanotubes (CNTs) with a diameter of about one hundred nanometres on a glass fibre membrane (GFM). The solar absorption of wet CNT-coated GFM can reach ~96% (Fig. 1c)

because of the porous fibrous light-trapping structure (Fig. 1d) and the inherent black property of the CNT[27]. Considering their abundant hydrophilic microchannels formed by intertwined glass fibres (Fig. 1e), the GFMs were also selected for use as MTBs. A GFM can immediately absorb a water droplet upon touching it because of its high affinity to water (Fig. 1f). Moreover, vertically aligned GFMs (i.e., MTBs) can pump water to 25 cm height in 60 min, demonstrating its strong capillary force for water transfer (Fig. 1g). A complete evaporation system was fabricated by assembling a number of MTBs and the solar absorber in a plastic frame (Fig. 1h, 1i and Fig. S1).

### Salt rejection capability

To avoid salt crystallization, excess salt must be efficiently transported back to maintain the top surface salinity below the saturation point. In this system, salt can be rejected via diffusion and convection through brine-filled microchannels under the driving force of the concentration gradient (osmosis) and gravity[25]. Its mass flow rate ($J$) can be described by the diffusion–convection equation as follows[28,29]:

$$J = J_{diff} + J_{conv} = nA\varepsilon(k_d(C_{evp} - C_0)/l + k_c(\rho_{evp} - \rho_0)) \qquad (1)$$

where $J_{diff}$ and $J_{conv}$ are the mass flow rate caused by diffusion and convection, respectively; $n$ is the number of MTBs; $A$, $\varepsilon$, and $l$ are the cross-section area, porosity, and height of the MTBs, respectively; $k_d$ and $k_c$ are the diffusion and average convective coefficients of salt, respectively; $C_{evp}$ and $C_0$ are the salt concentrations on the evaporation surface and in the bulk saline water, respectively; and $\rho_{evp}$ and $\rho_0$ are the salt solution densities on the evaporation surface and in the bulk saline water, respectively.

In Eq. (1), the mass transport rate is proportional to the bridge number $n$. We validated this relation by fabricating MTB structures with different bridge numbers ranging from 2 to 32 [Fig. 2a, cross-section area ($A$): ~0.135 cm²; height ($l$): 3 cm; porosity ($\varepsilon$): ~65%] and evaluating their evaporation performance using high-salinity water (10 wt.% NaCl). The evaluation was performed under 1 sun illumination for 12 h. Figure 2b shows that salt crystals massively accumulated on the 2-bridge evaporator surface because of insufficient mass transfer. This salt accumulation was mitigated with increase in the bridge number. For the evaporator containing 32 MTBs, no salt crystals were observed on the surface after the 12 h operation (Fig. 2b). At an insufficient number of MTBs (≤16), the evaporation rate gradually decreased as the vapour generation progressed because of the increased evaporation surface salinity (Fig. 2c, see the corresponding mass change curves in Fig. S2). In contrast, with sufficient MTBs (e.g., 32 bridges), the excess salt can be efficiently rejected to maintain the evaporation surface at a relatively low salinity. Remarkably, the evaporation rate of the 32-bridge evaporator was ~1.44 kg/m²/h without degradation during the 12 h operation.

Subsequently, we performed a complementary experiment to more intuitively demonstrate the salt backflow introduced by the 32-bridge evaporator. In this experiment, the evaporator was placed in a high-concentration saline water (10 wt.% NaCl solution) and exposed to 1 sun illumination, and 1 g of NaCl salt was added on its surface (upper panel, Fig. 2d). It was seen that during vapour generation, the added salt was gradually dissolved and completely removed in 11 h (lower panel, Fig. 2d; more details in Fig. S3). This experiment demonstrated that the salt backflow rate of the 32-bridge evaporator in the 10 wt.% NaCl solution was higher than the salt generation rate, thus confirming the salt rejection feature of the proposed MTB architecture. We further increased the brine salinity to test the maximum applicable salt concentration of this evaporator. Because the effects of diffusion and convection backflow decreased as the salinity (i.e., $C_0$ and $\rho_0$) increased, salt started to crystallize at the edges of the solar absorber after 12 h operation when 14 wt.% NaCl solution was used for the test (Fig. S4). Based on the corresponding evaporation rate, the salt

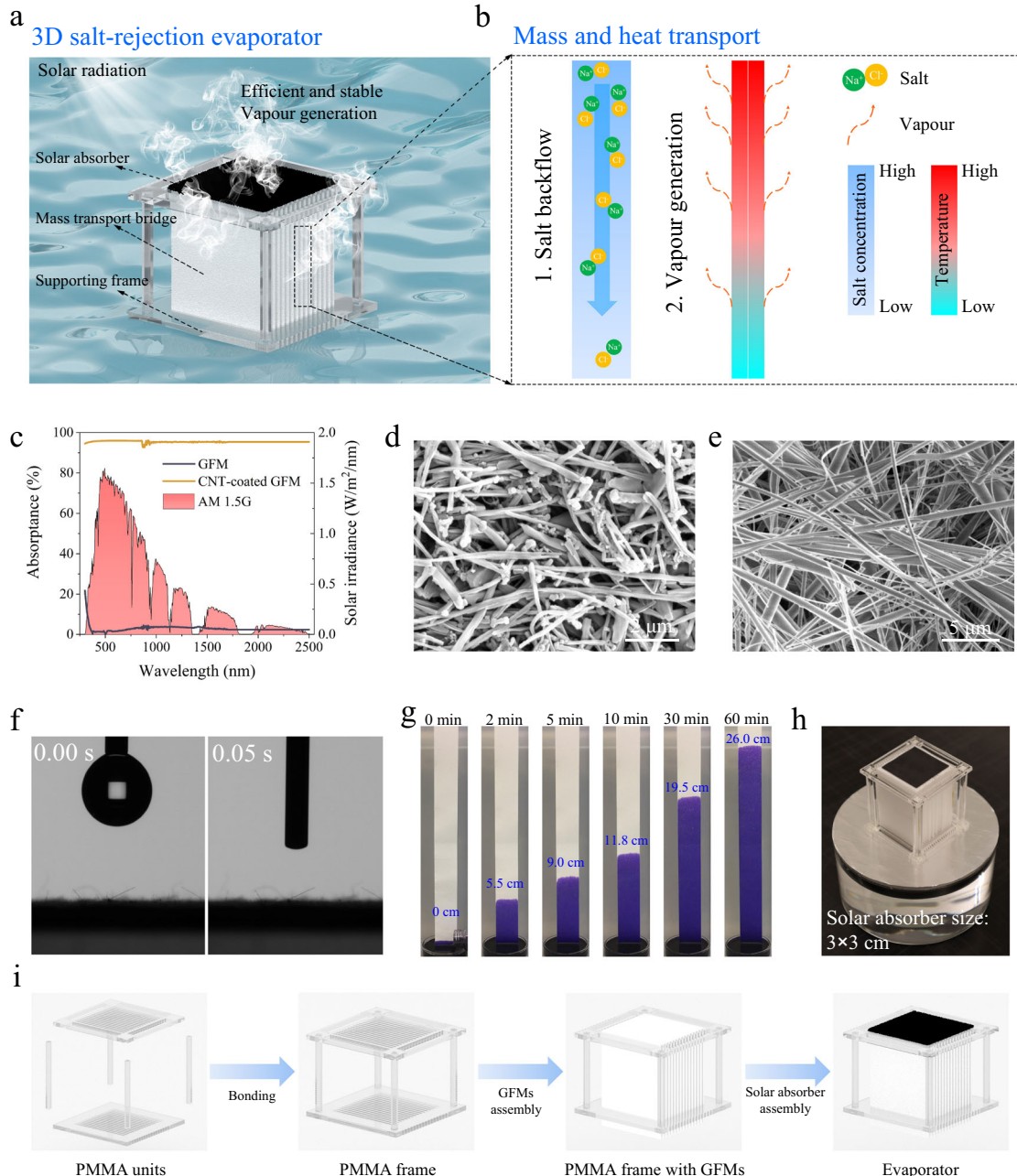

**Fig. 1 | Design and fabrication of the 3D salt-rejection evaporation structure.** **a** Schematic of the 3D salt-rejection solar evaporator. **b** Working principle includes salt rejection and evaporation enhancement. **c** UV–Vis–NIR spectra of the GFM, CNT-coated GFM, and standard solar irradiation spectrum of AM 1.5 G. **d** SEM image of the CNT-coated GFM surface. **e** SEM image of the GFM. **f** Image of the water drop hanging above the GFM and the moment it touches the GFM surface. **g** Anti-gravity transport of water along a GFM. **h** 3D salt-rejection evaporator prototype. **i** Schematic illustration of the fabricating process of the evaporator.

backflow along the MTBs was calculated as ~1.1 g/cm²/h. Interestingly, this unique mass transport feature is intertwined with its heat transport feature, as demonstrated in the subsequent section.

## Heat management

We fabricated 32-bridge evaporators with different bridge heights (Fig. 3a) and evaluated their evaporation performance. Under dark conditions, the evaporator without MTBs (i.e., bridge height: 0 cm) exhibited a natural evaporation rate of 0.15 kg/m²/h, which became more pronounced with the incorporation of MTBs due to the increased surface area (Fig. S5). Specifically, it linearly increased by ~0.04 kg/m²/h for every 1 cm increase in the MTB height. Under 1 sun illumination, the evaporation rate of the evaporator without MTBs was only 0.99 kg/m²/h because of the massive conductive heat dissipation

to the bulk water (Fig. 3b, see the mass change curves in Fig. S6). The MTB usage considerably promoted solar evaporation. The evaporation rate increased to 1.58–1.73 kg/m²/h when the bridge height reached 2–5 cm (Fig. 3b). These values are even higher than the theoretical upper limit for solar evaporation (~1.44 kg/m²/h, Supplementary Note 1 and Fig. S7), which can be attributed to the natural evaporation contribution (Fig. S8). When the MTB height exceeded 3 cm, the evaporation rate increased by ~0.04 kg/m²/h for every 1 cm increase in MTB height (Fig. 3b), which was consistent with the result obtained under dark conditions. This consistency suggests that the 3 cm height is sufficient for the MTB structure to maximize solar evaporation (note that additional increase in MTB height only increases natural evaporation). To reveal the mechanism of this observation, we analyzed the heat transport in this unique architecture.

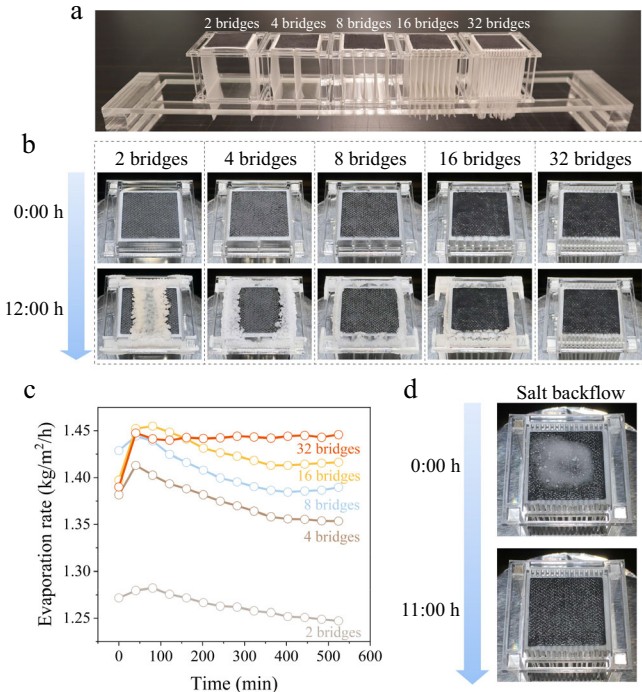

**Fig. 2 | Salt rejection performance. a** Photograph of evaporators with various bridge numbers (bridg height: 3 cm). **b** Photographic recordings of the salt accumulation on the 3D evaporators with different MTB numbers. **c** Evaporation rate variations of evaporators during long-term operations. **d** Photos of salt redissolving from the surface of a 32-bridge evaporator.

The energy loss channels for this evaporation system primarily include conductive heat loss into the bulk water ($P_{cond.}$), radiative heat loss ($P_{rad.}$), and convective heat loss to the environment ($P_{convec.}$). Therefore, the power flux available for evaporation ($P_{evp}$) can be described as follows[16]:

$$P_{evp} = P_{solar} - P_{cond.} - P_{rad.} - P_{convec.} \qquad (2)$$

where the solar energy input $P_{solar} = \alpha C_{opt} q_i$; $\alpha$ is the light absorption coefficient; $C_{opt}$ is the optical concentration; and $q_i$ is the direct solar illumination. The conductive heat flux $P_{cond.} = k(T_{sa} - T_{bw})/l$, where $k$ is the thermal conductivity; $T_{sa}$ and $T_{bw}$ are the temperatures of the solar absorber and the bulk water, respectively; and $l$ is the heat conduction path referring to the MTB height in our model. The radiative heat flux $P_{rad.} = \varepsilon \sigma (T_1^4 - T_2^4)$, while the convective heat flux $P_{convec.} = h(T_1 - T_2)$, $\varepsilon$ is the optical emission, $\sigma$ is the Stefan–Boltzmann constant, $h$ is the convection heat transfer coefficient, and $T_1$ and $T_2$ are the temperatures of the evaporator and environment, respectively.

The energy loss caused by the heat transfer from the top surface to the bulk water (i.e., $P_{cond.}$) can be minimized by increasing the MTB height (i.e., $l$) to confine the conductive heat within the MTB structure. This effect was visualized using infrared imaging to display the temperature gradients along MTBs with different heights. The results showed that the temperature at the bottom of the evaporator was similar to the ambient temperature when the MTB height reached or exceeded 3 cm (Fig. S9). This temperature distribution agreed well with the simulation modelled by COMSOL (Fig. S10). To obtain more insights into the heat transport in the architecture, we carefully recorded the internal temperature variation at different distances to the solar absorber under solar illumination. The results showed that the temperature stabilized after 60 min, when the internal temperature at 3 cm to the solar absorber was similar to that of the surrounding

environment (Fig. 3c), indicating that the conductive heat was completely confined in the top 3 cm of the MTB structure. This confinement effect was also demonstrated by the temperature change of the bulk water (Fig. 3d): for the evaporator without MTBs, the bulk water temperature increased from ~21 to ~26.2 °C after a 3 h operation due to the continuous heat input (left panel); for the evaporator with 3-cm MTBs, however, the bulk water temperature was maintained at room temperature (~21.3 °C) (right panel), thus confirming the suppression of heat dissipation into the bulk water.

Importantly, the confined heat energy can be exploited to generate additional vapour from the MTB surfaces, which can be efficiently released via the highly open interbridge spaces. To reveal this additional vapour generation from the vertical surfaces of MTBs, we used an evaporator having 32 MTBs (3 cm high) to perform a control experiment. In this experiment, the evaporator body was enclosed with an airtight polypropylene film, thus leaving only the upper surface exposed to the open space for vapour release (Fig. S11). After a 3 h operation, many water droplets condensed on the inner film surface, thus confirming that the MTBs released vapour (Fig. 3e). Compared to the completely open evaporator, the evaporation rate of the partially enclosed system decreased by ~31% (Fig. S12), demonstrating the importance of the open-channel design for enhanced interfacial evaporation.

Furthermore, we performed a cycling experiment to evaluate the evaporator stability. In each cycle, the evaporator ran for 12 h under 1 sun illumination and in a dark environment for another 12 h to simulate day and night alternation. Figure 3f shows that during this long-term test (with 10 wt.% NaCl solution), the mass change of the NaCl solution in each cycle linearly evolved and the evaporation rate stabilized at ~1.44 kg/m²/h. No performance degradation was observed after a seven-day cycling experiment.

Compared with the previously reported salt-rejection evaporators (evaporation rate: from 1.24 to 1.28 kg/m²/h for 10 wt.% NaCl solution)[9,26,30], our evaporator demonstrated a higher evaporation rate under similar conditions due to the heat confinement effect and the natural evaporation contribution. However, high evaporation efficiency alone is not sufficient for water production applications. If the evaporated moisture is not collected, it can only be considered as a pollutant to the environment considering that it has the greatest greenhouse effect among various components in the atmosphere[31]. Water collection that is equally important as vapour generation has been largely ignored in many previous studies on salt-rejection evaporators.

Therefore, we enclosed the evaporator with a transparent cover made of polymethyl methacrylate (PMMA) plates, creating a system that can produce water by condensing the evaporated moisture, and investigated the effects of bridge number and bridge height on the water production capacity of this system (Fig. S13a). When the bridge height was fixed at 3 cm, the amount of collected water increased with the number of bridges (Fig. S13b), which is consistent with the observation in the open system, confirming that the enhanced salt backflow facilitates water evaporation. When the bridge number was fixed at 32, the amount of collected water increased with the bridge height and reached the maximum at 3 cm, while further increasing the bridge height did not produce more water (Fig. S13c). This result is consistent with the conclusion above that 3 cm is sufficient to confine the conductive heat while further increasing bridge height only increases natural evaporation that does not contribute to water production. According to the three-hour test results, the water production rate of the enclosed evaporator in the optimal configuration (32 bridges; 3 cm high) is calculated to ~0.68 kg/m²/h (Fig. S13).

We also investigated the water generation performance of the enclosed system under different salinity conditions using NaCl solutions (3.5–20 wt.%). The results showed that the water production efficiency monotonically decreased from ~0.73 kg/m²/h for 3.5 wt.%

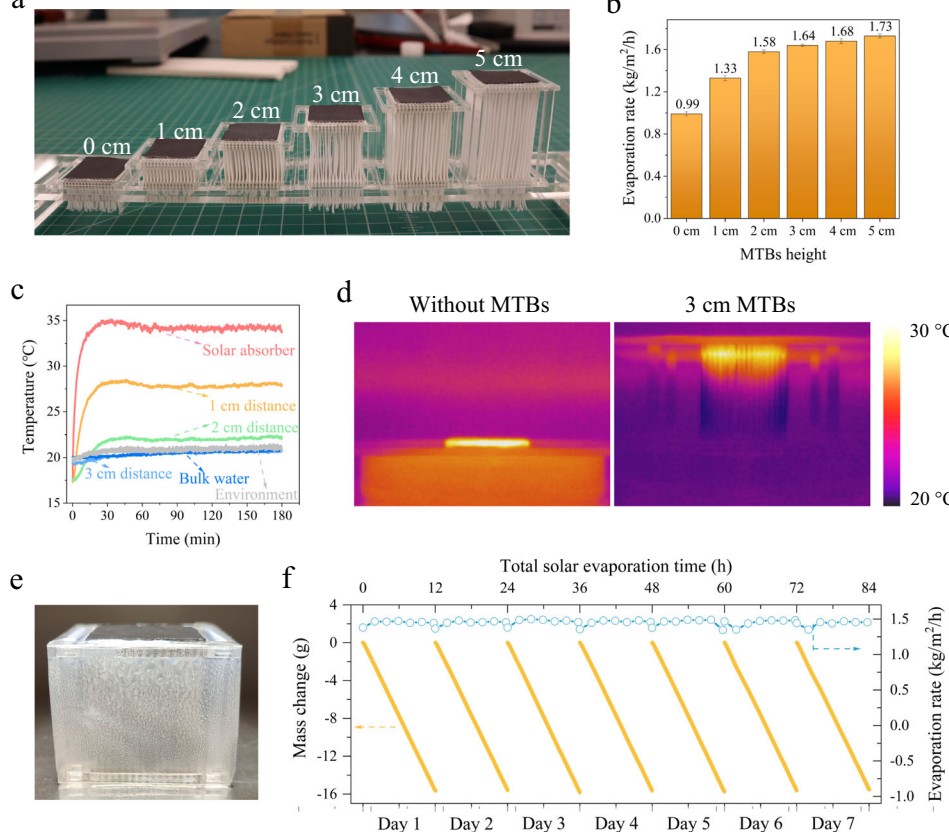

**Fig. 3 | Evaporation performance, heat management, and stability evaluation.**
**a** Photograph of evaporators with various bridge heights (bridge number: 32).
**b** Evaporation rate of evaporators with different MTB heights under 1 sun illumination (error bar type: standard deviation). **c** Internal temperature variation at different distances from the solar absorber. **d** Demonstration of the bulk water temperature after 3 h operation with different evaporators. **e** Photograph of the enclosed evaporator after 3 h evaporation. **f** Mass change curves and evaporation rate during the cycling experiment.

NaCl solution to ~0.63 kg/m²/h for 20 wt.% NaCl solution (Fig. S14a). The relatively low water production efficiency associated with the high-salinity brines is mainly due to their low saturated vapour pressure, partly due to the decreased photothermic conversion efficiency caused by salt precipitation. For instance, when using brine containing 20 wt.% NaCl, salt precipitation emerged at the periphery of the evaporator after three hours of testing (Fig. S14b).

### Field tests
As per the recently announced "best practice for solar water production"[32], the daily water yield is an important evaluation criterion that deserves additional consideration in practical implementations. Therefore, we prepared closed system based on the MTB structure and measured their water generation capacity under practical outdoor conditions.

**Rooftop experiment.** The fabricated solar-driven water generation system has a 15 × 26 cm² evaporator area (see Fig. S15). We first tested the system on the rooftop in KAUST, Thuwal, Saudi Arabia (Fig. S16). In this experiment, we employed the discharged water from an RO system of the KAUST Seawater Desalination Plant as the source water (salinity: ~8.7%). Our daily evaluation started at 8:00 and ended at 17:00. As shown in Fig. 4a, the evaporator surface was heated by solar light to a temperature 4–15 °C higher than the environment. However, the temperature at the bridge bottom was almost the same as the environment temperature, indicating that the conductive heat was confined, with only a small amount transferred to the bulk water. Consequently, saline water can be efficiently evaporated and condensed at the cover surface for the water collection. Figure 4b and

Supplementary Movie 1 illustrate the relevant details. The total collected water was ~175 ml, of which ~110 ml flowed in the graduated cylinder, and ~65 ml was retained in the PMMA cover. Based on the evaporator area (390 cm²), the daily water productivity was calculated as ~5.0 L/m². We measured the ion contentions of our water samples to evaluate the water quality. Compared with the discharged water from the RO plant, the ion concentration of condensed water was reduced by at least four orders of magnitude, thus fully meeting the WHO drinking water requirements (Fig. 4c).

We calculated its practical solar–water collection efficiency of the system, $\eta_{prac}$, using Eq. (3):

$$\eta_{prac} = m_{cond}h_{lv} / \left( A_{evp} \int q_{solar}(t)dt \right) \qquad (3)$$

where $m_{cond}$ is the daily water collection amount; $h_{lv}$ is the total enthalpy of the liquid–vapour phase transition; $A_{evp}$ is the evaporator area; and $q_{solar}$ is the time-dependent solar flux. Benefiting from the highly efficient vapour generation, the overall solar–water collection efficiency of our system reached ~41.6%, representing a considerable improvement compared to the previously reported salt-rejection solar evaporation systems (e.g., maximum efficiency of a rooftop system: ~24%[25]). We performed a continuous test from Apr. 7 to Apr. 11, 2022 to evaluate the performance stability (Fig. 4d). The daily water collection rate fluctuated in the range of 4.7–5.2 L/m² depending on the specific solar insolation of the day. The corresponding solar–water collection efficiency was 39%–42%. Remarkably, no salt accumulation was observed during this five-day outdoor operation (Fig. 4e). These results demonstrate the potential of the fabricated

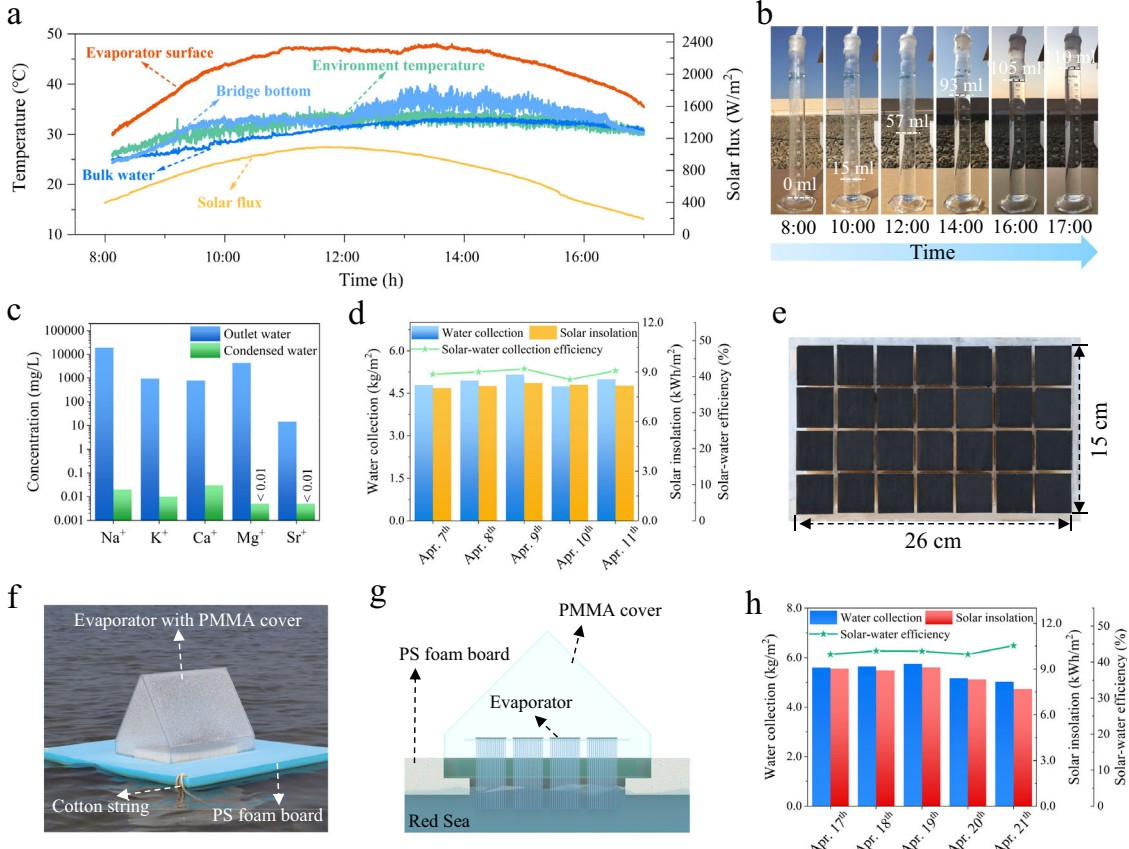

**Fig. 4 | Field tests. a** Real-time temperature variation of the solar absorber, environment, bottom of bridges and bulk water, and solar flux from 8:00 to 17:00 on Apr. 11, 2022. **b** Timelapse photos of the collected water in the graduated cylinder from 8:00 to 17:00. **c** Ion concentration in the effluent water collected from the RO facility and collected freshwater from our system. **d** Daily water generation, solar insolation, and solar–water collection efficiency from Apr. 7 to 11, 2022. **e** Photograph of the evaporator after five-day operation. **f** Photograph of the floating system for the ocean test. **g** Schematic illustration of the structure of the floating system. **h** Daily water collection, solar insolation, and solar–water efficiency during the ocean test from Apr. 17 to 21, 2022.

evaporator to extract freshwater from the wastewater discharged by RO plants.

**Floating test**. After the 5-day rooftop experiment, the same MTB-based evaporation system was tested in a floating configuration in the Red Sea (salt content: ~4.3%) to demonstrate its potential for practical seawater desalination (Fig. 4f, g). The test started and ended at 8:00 and 17:00, respectively, each day and lasted for five days from Apr. 17 to Apr. 21, 2022. As shown in Fig. 4h, the daily freshwater productivity ranged from 5.0 to 5.8 L/m$^2$ with a stable solar–water collection efficiency of 42%–45%, which was consistent with the rooftop test. This freshwater productivity was approximately two times higher than the previous record of the salt-rejection solar evaporator (~2.5 L/m$^2$ per day)[25]. The field test demonstrated a high-performance solar evaporator that will help in disaster relief or strengthen the resilience of individuals living on boats and coastal areas.

## Discussion

In summary, we designed a solar evaporator architecture to simultaneously achieve excellent salt rejection and high water evaporation, thus realizing stable and efficient freshwater production from various types of brines. The major factor for the success of this system was the usage of rationally designed MTBs that not only facilitated salt and water transport but also recovered conductive heat to evaporate water. This design allowed water evaporation to occur in three dimensions rather than being confined to the evaporator surface as in conventional salt-rejection systems; hence, the overall efficiency was

significantly improved. The fabricated evaporator can continuously and stably work under solar light to extract freshwater from contaminated or saline water, including, but not limited to, seawater or concentrated brine from RO facilities. The carefully designed indoor and outdoor experiments demonstrated an improved solar-to-water collection efficiency of >40%, with a daily water generation of ~5.0 L/m$^2$ under natural sunlight illumination. This means that a one-square-metre evaporator can meet the drinking needs of more than two people, according to the recommendation from the European Food Safety Authority that the daily drinking water requirements for females and males are 2.0 L/day and 2.5 L/day, respectively[33]. The water production performance of the evaporator remained unchanged during the 10 days of outdoor tests, demonstrating the good stability required for practical applications. Based on the current laboratory scale manufacturing, the fabrication cost of our solar evaporator is around $45/m$^2$ (see Supplementary Note 2). While such a cost is not low, it is expected to be significantly reduced in mass production, especially when low-cost alternative materials are used. Because of its high evaporation rate, salt-rejection capability, and scalable manufacturability, this 3D evaporator holds considerable promise for freshwater supply, particularly for those living in water-stressed coastal areas and off-grid regions.

## Methods
### Evaporator fabrication
The solar absorber was fabricated by loading partially oxidized CNTs on the GFM. The reason for choosing partially oxidized CNTs over

pristine CNTs is that their hydrophilicity facilitates the formation of a uniform coating on the GFM. First, 3 g CNTs (diameter: 110–170 nm and length: 5–9 μm, Sigma-Aldrich) was dispersed in a 120 ml acid mixture (90 ml $H_2SO_4$ + 30 ml $HNO_3$) and reacted at 70 °C for 5 h. The resultant product was collected by filtration and washed until neutralization. Second, a certain amount of partially oxidized CNTs was dispersed in water through ultrasonication and then filtered through a GFM. The obtained solar absorber was dried at 60 °C. The CNT loading percentage was determined to be ~11 wt.%. The MTBs were composed of GFM. The GFM sheets were 0.45 mm thick and 3 cm wide. The supporting plastic frame was prepared using a PMMA plate via laser cutting and bonding. The final evaporator structure was developed by assembling the GFM sheets (width: 3 cm and thickness: 0.45 mm), the solar absorber, and the PMMA frame.

### Structure characterization
The GFM and CNT-coated GFM microstructures were characterized by SEM (Teneo VS, FEI). Their light absorption spectra were obtained under the transmission and reflection mode using a UV/Vis/NIR spectrometer (LAMBDA 950, PerkinElmer). The porosity was determined using a mercury porosimeter (AutoPore V, Micromeritics). The cation concentration was measured by ICP-OES (5110 ICP-OES, Agilent). The IR images of the evaporator were captured by an IR camera (H16, Hikvision).

### Performance evaluation
The evaporation performance was performed with a home-made solar evaporation system equipped with a solar simulator (91160-1000, Newport) to generate solar light and a microbalance to record the weight change. The relative humidity and room temperature of the lab were ~60% and 21 °C, respectively. For outdoor experiments, the solar intensity was measured using a photometer (TM-208, Tenmars), and the temperature variation was recorded by a multi-channel thermometer (JK808, Jinko). The water collection process was then recorded by video.

## Data availability
All data that support the findings of this study are available in the paper or its Supplementary Information. Source data are provided with this paper.

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

## Acknowledgements
This work is supported by KAUST baseline Fund to Y.H. and the National Key Research and Development Project of China (2022YFE0113800). The authors thank Mohammad I. Hussain and Amal Thundiyil Ambili Chandran from the FM team for providing the discharged water from the RO system of the KAUST seawater desalination plant.

## Author contributions
K.Y. and T.P. designed the experiments and carried out this project. S.D. performed the COMSOL simulation. Q.G and Y.H. designed and supervised this project, and revised the manuscript.

## Competing interests
The authors declare no competing interests.
