## [Peer Review File · Nature Communications]

Three-dimensional open architecture enabling salt-rejection solar evaporators with boosted water production efficiencyREVIEWER COMMENTS

Reviewer #1 (Remarks to the Author):

The paper entitled "Three-dimensional open architecture enabling salt-rejection solar evaporators with boosted water production efficiency" proposes a novel solution to limit salt crystal formation during solar water evaporation. This work brings very interesting research to this field and deserves to be published. The article is well written and I did not find any problem with the form. In my opinion the figures S1 S2 and S15 should be integrated in the article, otherwise it is difficult to understand what it is about.

Below are two remarks to improve the substance of the article:

In the nighttime conditions one can clearly see the impact of the exchange area with the increase in the number of bridges. The problem of the study is that the performances are calculated in relation to the surface of the evaporator. However, increasing the number of bridges also increases the evaporator surface. In order to limit the effect of the exchange surface, it would have been interesting to estimate the evaporation performances for different heights of MTBs and for a variable number of MTBs in the case where the evaporator is enclosed. Such a study would increase the quality of the paper.

On page 5, it is explained that the salt crystallizes at 14% concentration. What is the performance of the evaporator in this case. It would have been interesting to make a study according to the salinity of the water (e.g. 5, 10 15 20%) for several heights and several MTBs in a systematic way and for a case where the evaporator is closed. By creating 3D graphs, it would be possible to know under which conditions the performances degrade.

Reviewer #2 (Remarks to the Author):

The manuscript 'Three-dimensional open architecture enabling salt-rejection solar evaporators with boosted water production efficiency' investigates vertically aligned mass transfer bridges in the solar evaporator to produce clean water. Although it is quite interesting, it's not ready to publish in the respected journal yet. Please refer to the following comments, which are helpful to the authors.

1. Line numbers 44 and 62: Although hydrophobic properties can prevent salt accumulation on the solar-absorber layer, a high concentration of saline water can penetrate that layer due to a low surface free energy of the high concentration of saline water. Additionally, if the authors emphasize the novelty, the authors should add some more data related to the contact angle of different NaCl concentration droplets on the absorber.
2. Line numbers 96 to 98: If this process is applied in a pilot or real scale levels, it takes more time than this experiment. Does this process has the feasibility to produce clean water at a commercial level?
3. Line numbers 120 to 121: A 12 h operation is insufficient to conduct occurring salt crystals.
4. Line numbers 131 to 133: The reviewer thinks that 1 g of NaCl is dissolved and penetrated to the opposite side due to a reverse salt flux due to the osmosis. The authors should confirm a mechanism to dissolve 1 g of NaCl.
5. Line numbers 137 to 139: It's related to the solubility of NaCl. A 14 wt% NaCl should be supersaturated on the absorber.
6. Line numbers 144 to 160: Although it seemed to be a significant experiment, it doesn't have a meaning in terms of feasibility. For example, if the authors change material instead of CNTs, it should be totally different.
7. Line number 232: It exhibited 5 L/m²/d, which means that it showed 0.56 L/m²/h. Is it possible to make a commercial level? The process is still insufficient to produce clean water instead of a commercial level of RO brine treatment processes such as membrane distillation in terms of

feasibility.

8. Line numbers 250 to 251: If the evaporator will treat RO brine (e.g., RO brine of 1,000 t/d), how about the footprint of the evaporator?

9. Line numbers 277 to 278: Why are the authors applying partially oxidized CNTs on the GFM? Although it has a high conductive heat, the authors should explain it in the manuscript. Additionally, what is the difference between CNTs compared to the previous study?

Point-by-Point Responses to the Reviewers' Comments

Reviewer: 1

Comments: The paper entitled "Three-dimensional open architecture enabling salt-rejection solar evaporators with boosted water production efficiency" proposes a novel solution to limit salt crystal formation during solar water evaporation. This work brings very interesting research to this field and deserves to be published. The article is well written and I did not find any problem with the form. In my opinion the figures S1 S2 and S15 should be integrated in the article, otherwise it is difficult to understand what it is about.

Responses: We are grateful to the reviewer for her/his appreciation of our work. Following the reviewer's suggestion, we have integrated Fig. S1, S2, S6 and S15 into Fig. 1, 2, 3 and 4 in the revised manuscript, respectively.

Comments: In the nighttime conditions one can clearly see the impact of the exchange area with the increase in the number of bridges. The problem of the study is that the performances are calculated in relation to the surface of the evaporator. However, increasing the number of bridges also increases the evaporator surface. In order to limit the effect of the exchange surface, it would have been interesting to estimate the evaporation performances for different heights of MTBs and for a variable number of MTBs in the case where the evaporator is enclosed. Such a study would increase the quality of the paper.

Responses: We thank the reviewer for this insightful comment. The reviewer is correct that the effect of exchange surface (i.e., the contribution of natural evaporation) can be largely eliminated by using an enclosed system for the measurement. In fact, the system we used for field tests was closed for achieving water collection. The reason we discuss open systems in the manuscript is for easier comparison with literature results on water evaporation (most previous works did not discuss water collection).

As suggested by the reviewer, we have added one paragraph to the revised manuscript discussing the effects of bridge number and bridge height on the water generation capacity of the enclosed evaporator (Fig. R1a). When the bridge height was fixed at 3 cm, the amount of collected water increased with the number of bridges (Fig. R1b). This result can be attributed to the alleviated salt accumulation and the decrease in salt concentration on the evaporating surface. When the bridge number was fixed at 32, the amount of collected water increased with the bridge height and reached the maximum at 3 cm, while further increasing the bridge height did not produce more water (Fig. R1c). This result is consistent with the conclusion above that 3 cm is sufficient to confine the conductive heat while further increasing bridge height only increases natural evaporation that has no effect on water

collection. In the revised manuscript, the new data for the enclosed system are shown in Fig. S13 and discussed in the main text.

Figure R1. Water generation using an enclosed solar evaporation system. (a) Photograph of the enclosed solar evaporation system for water collection. (b) The effect of the number of bridges on water generation, where the bridge height is fixed at 3 cm. (c) The effect of the height of bridges on water generation, where the bridge number is fixed at 32. The experiments were performed with 10 wt% NaCl solution under 1 sun illumination. In (b), each test was performed for 12 hours. In (c), each test was performed for 3 hours.

Comments: On page 5, it is explained that the salt crystallizes at 14% concentration. What is the performance of the evaporator in this case? It would have been interesting to make a study according to the salinity of the water (e.g. 5, 10 15 20%) for several heights and several MTBs in a systematic way and for a case where the evaporator is closed. By creating 3D graphs, it would be possible to know under which conditions the performances degrade.

Responses: We thank the reviewer for raising this point. Following the reviewer's suggestion, we discuss the water generation performance of an enclosed system under different salinity conditions in the revised manuscript. Given that the effects of bridge number and bridge height have been thoroughly investigated, we used the optimized conditions (i.e., 32 bridges; 3 cm high) to study water generation from brines with different salinities (wt% of NaCl).

The results showed that the water production efficiency monotonically decreased from ~ 0.73 kg/m²/h for 3.5 wt% brine to ~ 0.63 kg/m²/h for 20 wt% brine (Fig. R2a). The relatively low water production efficiency associated with the high-salinity brines is mainly due to their low saturated vapor pressure, partly due to the decreased photothermic conversion efficiency caused by salt precipitation. For instance, when using brine containing 20 wt% NaCl, salt precipitation was observed at the periphery of the evaporator after three hours of testing (Fig. R2b). These results are shown in Fig. S14 and discussed in the main text in the revised manuscript.

Figure R2. (a) Water generation from brines of varying salinities (wt% of NaCl) using an enclosed solar evaporation system with an optimized MTB architecture. Each test was performed under 1 sun illumination for 3 hours. (b) Photograph of the evaporator after 3 h of operation in 20 wt% brine, showing salt precipitation emerged at the periphery of the evaporator surface.

Reviewer: 2

Comments: The manuscript 'Three-dimensional open architecture enabling salt-rejection solar evaporators with boosted water production efficiency' investigates vertically aligned mass transfer bridges in the solar evaporator to produce clean water. Although it is quite interesting, it's not ready to publish in the respected journal yet. Please refer to the following comments, which are helpful to the authors.

Responses: We sincerely thank the reviewer for the insightful comments that indeed helped improve the quality of the manuscript.

Comments: Line numbers 44 and 62: Although hydrophobic properties can prevent salt accumulation on the solar-absorber layer, a high concentration of saline water can penetrate that layer due to a low surface free energy of the high concentration of saline water. Additionally, if the authors emphasize the novelty, the authors should add some more data related to the contact angle of different NaCl concentration droplets on the absorber.

Responses: We agree with the reviewer that “a high concentration of saline water can penetrate the hydrophobic layer due to a low surface free energy”. However, the reviewer might have some misunderstandings about our work. In our study, instead of using a hydrophobic layer to prevent brine penetration, we intentionally chose hydrophilic glass fiber membranes (i.e., MTBs) to connect the brine and the evaporator surface. Because the MTBs have highly hydrophilic channels, they can absorb brine immediately upon contact. Therefore, no contact angle analysis has been conducted. In our system, “salt rejection” is achieved by salt backflow via the MTBs, not by using a hydrophobic layer.

Comments: Line numbers 96 to 98: If this process is applied in a pilot or real scale levels, it takes more time than this experiment. Does this process has the feasibility to produce clean water at a commercial level?

Responses: We thank the reviewer for raising this important concern. Our system can operate at larger scales by assembling the evaporator units into lateral arrays (see Fig. R3). Since the height of MTBs does not change, the time required to deliver brines to the evaporating surface will not change.

Figure R3. Schematic illustration of module construction for large-scale applications.

Comments: Line numbers 120 to 121: A 12 h operation is insufficient to conduct occurring salt crystals.

Responses: We thank the reviewer for raising this point. For solar evaporation application, the evaporator works in sunlight during the day and then goes into a self-cleaning process at night. Therefore, operation of 12 hour under 1 sun illumination is sufficient for evaluating the salt rejection ability of the evaporator, considering that there is typically less than 12 h of full sun per day. Moreover, we have demonstrated that the evaporator can operate continuously for many days without degrading the performance (Fig. 3f and Fig. 4), which is a strong testament to its salt rejection property. In addition, the extra NaCl added on the evaporating surface could be brought into the bulk brine (10 wt% NaCl solution) under 1 sun illumination (Fig. 2d), further confirming the salt rejection feature of this evaporator.

Comments: Line numbers 131 to 133: The reviewer thinks that 1 g of NaCl is dissolved and penetrated to the opposite side due to a reverse salt flux due to the osmosis. The authors should confirm a mechanism to dissolve 1 g of NaCl.

Responses: The reviewer is correct that the salt backflow is driven by osmosis. As the water evaporates, the salt concentration at the evaporating surface increases. Consequently, the salt can be transported downward via diffusion and convection, driven by the concentration gradient and gravity, respectively. This mechanism was verified in a previous study (*Energy Environ. Sci.*, **2018**, *11*, 1510-1519). Nevertheless, efficient salt backflow requires adequate water transport paths, otherwise salt can precipitate on the evaporating surface. The MTB structure of our evaporator significantly facilitates water transport. In the revised manuscript, we have clarified the mechanism in the section of *salt rejection capability* and cited this reference, following the reviewer's suggestion.

Comments: Line numbers 137 to 139: It's related to the solubility of NaCl. A 14 wt% NaCl should be supersaturated on the absorber.

Responses: The saturated concentration of NaCl in water is ~26 wt% at 25°C and it increases with the temperature (*Chem. Eng. Data*, **2005**, *50*, 29–32). Under our experimental conditions, the NaCl solution (14 wt%) is not supersaturated.

Comments: Line numbers 144 to 160: Although it seemed to be a significant experiment, it doesn't have a meaning in terms of feasibility. For example, if the authors change material instead of CNTs, it should be totally different.

Responses: The true value of our study lies in the unique design of solar evaporator architecture, which allows the full use of conductive heat to generate vapor in three dimensions. With this design, the material used to fabricate the evaporating surface is not limited to CNT. Any materials with good solar-thermal efficiencies are suitable candidates, such as graphene, carbon black powders and carbon fibers. In our study, CNT was used simply because of its easy availability.

Moreover, the heat conduction along the MTBs is also unrelated to the CNTs located on the top surface of the evaporator. Therefore, changing CNTs to other carbon materials will not substantially change the overall water production performance of the system. For instance, in our previous publication (*Global challenge*, **2017**, *1*, 1600003.), carbon black powders were used for photo-thermal energy conversion. The reported evaporation rate was also excellent, up to 1.28 kg/m²/h.

Comments: Line number 232: It exhibited 5 L/m²/d, which means that it showed 0.56 L/m²/h. Is it possible to make a commercial level? The process is still insufficient to produce clean water instead of a commercial level of RO brine treatment processes such as membrane distillation in terms of feasibility.

Responses: We appreciate this comment. Solar evaporation technology is not developed to replace the traditional RO desalination technologies. Instead, this technology is used for disaster relief or off-grid operations in less-developed areas where the traditional desalination technology is not applicable.

According to the European Food Safety Authority, the daily drinking water requirements for females and males are 2.0 L/day and 2.5 L/day, respectively. Therefore, 5 L/m²/d water production is sufficient to satisfy the daily drinking water requirement for individuals. It is worth noting that this water productivity has outperformed that of the commercial solar still (Aquamate Solar Still®) that is developed for the same purposes. In the revised manuscript, we have further clarified this point.

Comments: Line numbers 250 to 251: If the evaporator will treat RO brine (e.g., RO brine of 1,000 t/d), how about the footprint of the evaporator?

Responses: We thank the reviewer for asking this insightful question. Given that solar water evaporation is a zero carbon footprint technology, we presume that the reviewer asked about the land footprint.

Currently, using evaporation ponds is an option to treat RO brines to achieve zero liquid discharge. Compared with the evaporation of open water surfaces under 1 sun illumination (~0.40 kg/m²), the evaporation rate of our system is more than 4 times higher (~1.64 kg/m²), corresponding to a 4-fold reduction in required area. In addition, our 3D evaporator exhibits a higher evaporation rate than the cutting-edge salt-rejection evaporators (*Energy Environ. Sci.*, **2021**, *14*, 5347-5357; *Nat. Commun.*, **2022**, *13*, 849; *Sci. Adv.*, **2019**, *5*, eaaw7013), which has been discussed in our manuscript.

Comments: Line numbers 277 to 278: Why are the authors applying partially oxidized CNTs on the GFM? Although it has a high conductive heat, the authors should explain it in the manuscript. Additionally, what is the difference between CNTs compared to the previous study?

Responses: We thank the reviewer for raising this point. The reason for choosing partially oxidized CNTs over pristine CNTs to fabricate the evaporating surface is the poor affinity of pristine CNTs to the hydrophilic glass fiber membrane (GFM). Partial oxidation of CNTs makes them more hydrophilic and thus easier to form a uniform coating on the GFM. The CNTs we used are commercially available with no noticeable differences from those used in previous studies. In the revised manuscript, we have explained the reason for using partially oxidized CNTs, following the reviewer's suggestion.

REVIEWERS' COMMENTS

Reviewer #1 (Remarks to the Author):

The authors have responded correctly to my comments and have completed the manuscript accordingly. Therefore, I give my consent for this article to be published.

Reviewer #2 (Remarks to the Author):

The revised manuscript should be accepted in your respected journal.